# Epigenetic Mechanisms of Postoperative Cognitive Impairment Induced by Anesthesia and Neuroinflammation

**DOI:** 10.3390/cells11192954

**Published:** 2022-09-21

**Authors:** Katharina Rump, Michael Adamzik

**Affiliations:** Department of Anesthesiology, Intensive Care Medicine and Pain Therapy, University Hospital Knappschaftskrankenhaus Bochum, Ruhr-University Bochum, In der Schornau 23–25, 44892 Bochum, Germany

**Keywords:** epigenetics, anesthesia, BDNF, inflammation, neuroinflammation, DNMTs, HDACs, postoperative cognitive dysfunction, postoperative delirium

## Abstract

Cognitive impairment after surgery is a common problem, affects mainly the elderly, and can be divided into postoperative delirium and postoperative cognitive dysfunction. Both phenomena are accompanied by neuroinflammation; however, the precise molecular mechanisms underlying cognitive impairment after anesthesia are not yet fully understood. Anesthesiological drugs can have a longer-term influence on protein transcription, thus, epigenetics is a possible mechanism that impacts on cognitive function. Epigenetic mechanisms may be responsible for long-lasting effects and may implicate novel therapeutic approaches. Hence, we here summarize the existing literature connecting postoperative cognitive impairment to anesthesia. It becomes clear that anesthetics alter the expression of DNA and histone modifying enzymes, which, in turn, affect epigenetic markers, such as methylation, histone acetylation and histone methylation on inflammatory genes (e.g., TNF-alpha, IL-6 or IL1 beta) and genes which are responsible for neuronal development (such as brain-derived neurotrophic factor). Neuroinflammation is generally increased after anesthesia and neuronal growth decreased. All these changes can induce cognitive impairment. The inhibition of histone deacetylase especially alleviates cognitive impairment after surgery and might be a novel therapeutic option for treatment. However, further research with human subjects is necessary because most findings are from animal models.

## 1. Introduction

The first reports of long-lasting cognitive impairment in older patients after surgery were published more than six decades ago [1]. Despite intense research in the last few decades, little is known about effectively preventing perioperative stress in the aging brain [2]. Two major disorders are described in the context with postoperative cognitive impairment: postoperative delirium (POD) and postoperative cognitive dysfunction (POCD) [3,4]. Delirium, or acute confusional syndrome, is characterized by impaired orientation, diminished attention, and aberrant perception. It is defined as a syndrome characterized by fluctuating cognitive impairment of acute onset that resolves within days of a surgical procedure. The pathogenesis is multifactorial and it frequently appears in elderly patients admitted to general hospitals [5]. The contrast to cognitive impairment without confusional syndrome is that it varies in its duration. It can last 3 months or longer [6]. In this context we also have to mention mild cognitive impairment and dementia, which are not the same although both are characterized by objective evidence of cognitive impairment. The main difference is that in dementia, more than one cognitive domain is invariably involved and substantial interference with daily life is evident [7]. In addition, dementia is an irreversible state, which distinguishes it from the here described phenomena. The relationship between POD and POCD is not fully understood and overlap and divergence are currently being discussed [2,8]. Delirium may contribute to POCD or long-term cognitive impairment due to several reasons. One is that patients with delirium are often incapable of maintaining hydration and nourishment, even with good nursing care, in addition sleep deprivation might occur and cause neuronal damage. In addition drug toxicity might occur, an example being the use of benzodiazepines associated with cognitive impairment [9]. In general, POD is an acute brain failure, occurs directly after surgery in 15–53% of older patients and is defined by the Diagnostic and Statistical Manual of Mental Disorders and assessed by the Confusion Assessment Method for Intensive Care Unit [4,10,11] or by the Nursing Delirium Screening Scale [12]. It is associated with a longer stay in hospital, prolonged ventilation time, higher nursing effort and increased mortality [13]. Postoperative delirium recovers with high intraindividual differences from hours to months [14] and can appear as either a more frequent hypoactive subtype or a hyperactive or mixed form, which is accompanied by apathy, confusion, agitation or delusions, together with signs of sympathetic arousal [15]. In contrast to POD, POCD is not a clinical diagnosis but more a variably operationalized concept defined by a decrease in cognitive performance after surgery, as measured by neuropsychologic tests, and occurs in 5–55% of older patients within the first three postoperative months [16]. It manifests as a dysfunction of memory, mental capacity, language ability or other aspects of cerebral function [17]. It was demonstrated recently that POD increases the risk of POCD significantly in the first postoperative month, while this relationship does not seem to be stable in the longer-term follow-up [2]. The pathophysiology of the syndromes is still not well understood, but it is assumed that in delirium an acute change of neurotransmitter levels occur, while long term cognitive impairment is accompanied by chronic neurodegeneration and triggered by neurotoxicity [9]. Despite the fact that the pathophysiology of either disease is not fully understood, there seem to be similarities, as neuroinflammation plays a key role in both [18]. However other risk factors that may contribute to cognitive impairment must be mentioned. They include pre-operative general and cognitive health, medical comorbidities such as hypotension, hypoxia, hospital admission for any cause, exposure to general anesthesia and psychoactive drugs [19]. In the postsurgical period, neuroinflammation caused by surgery is considered a major contributor to the development and progress of atherosclerosis and neurodegenerative diseases that may lead to an increased systemic and cerebral inflammatory response to surgery. In addition postoperative complications, such as infection, pulmonary complications and hypoxemia may occur and aggravate the postoperative inflammatory reaction [20]. In addition, the usage of distinct anesthetics is associated with long-term morphological and functional alterations in the brain, especially when administered at the extremes of age [21,22]. However, as postoperative cognitive impairment can last over a long time period, it seems appropriate to speculate that one possible mechanism is an alteration of the epigenetic profile in the human brain or related cells. Hence, we review the available mechanistic data of epigenetic processes linking general anesthesia exposure and neuroinflammation to impaired cognitive performance. Accordingly, we first give a brief overview of the pathomechanisms of postoperative cognitive impairment, then summarize the epigenetic effects caused by neuroinflammation and anesthetics, and finally evaluate the current literature linking epigenetics to cognitive impairment.
**Main hypotheses of the review:**-Cognitive impairment is a common problem after surgery-Epigenetic dysregulation may play a role in its development-Anesthetics alter expression of DNA and histone-modifying enzymes-This causes an altered epigenetic profile of genes involved in neuroinflammation and neuronal maintenance and development-Epigenetic drugs may be a novel therapeutic option for the treatment of postoperative cognitive impairment

## 2. Molecular Mechanisms of Postoperative Cognitive Impairment

The pathophysiology of POD and POCD is closely related to neuroinflammation [23]. The biomarkers currently discussed display mainly a set of not only inflammatory markers [24], among them interleukin (IL-)6, C-reactive protein [25], IL-8, IL-10 and matrix metalloproteinase 8 [26], but also the brain-related markers S100 calcium-binding protein B and neuron-specific enolase [15]. In addition, markers belonging to the cholinergic system, such as acetylcholinesterase and butyrylcholinesterase, have been studied and assessed as possible biomarkers for delirium [27].

The current understanding of the development of POD and POCD is that surgery causes trauma of the surgical site accompanied by the release of damage-associated patterns [23,28] (Figure 1). The most prominent damage-associated patterns in this context are high mobility group protein B1 (HMGB1) and S100 calcium-binding protein A8 [29]. HMGB1 can bind to toll-like receptors TLR2 and TLR4 of different immune cells, and activates nuclear factor kappa B, which, in turn, causes the release of several cytokines [30]. The pro-inflammatory cytokines IL-1 beta (IL-1β), IL-6 and tumor necrosis factor alpha cause a further release of HMGB1 in a positive feedback loop, amplifying the inflammatory response and the upregulation of cyclooxygenase 2 isozyme release [31]. Circulating peripheral pro-inflammatory cytokines disrupt blood-brain barrier (BBB) permeability via cyclooxygenase 2 upregulation and MMPs, allowing pro-inflammatory cytokines to enter the central nervous system (CNS) [32]. After the disruption of the BBB, peripheral macrophages can enter the brain, triggered by monocyte chemoattractant protein 1, and can activate microglia, which are known as the resident macrophages of the CNS [33]. When microglia are activated, they themselves release cytokines, and neuroinflammation is amplified. In addition, microglia can release reactive oxygen species in response to HMGB1, which aggravates cognitive impairment [31,34]. Disruption of the BBB, in turn, leads to the release of CNS-specific proteins S100 calcium-binding protein B and neuron-specific enolase to the plasma [35] (Figure 1). In addition, microglia astrocytes are activated and promote inflammation and the release of glutamate and glycogen [17]. All these processes cause damage to several functions in the brain, including the depression of inhibitory neurotransmission via downregulation of gamma-aminobutyric acid receptors, disrupting the balance of excitatory and inhibitory neurotransmission and, ultimately, favoring glutamate toxicity [36], the pathological activation of astrocytes, which causes synaptic dysfunction [28], and the reduction of cerebral acetylcholine, triggering the complex neuroinflammatory response and the subsequent degeneration of cholinergic neurons through neural necrocytosis, apoptosis and pyroptosis [13]. In addition, the cholinergic anti-inflammatory pathway seems to play an important role in the pathophysiology. Surgical trauma induces activation of the sympathetic nervous system, and the vagus reflex is stimulated, which causes the release of acetylcholine and activates the nicotinic α7 receptor on macrophages [28], which causes the inactivation of the nuclear factor ‘kappa-light-chain-enhancer’ of activated B (NF-kB) cells and decreases cytokine release [37,38]. Hence, vagus nerve stimulation may attenuate postoperative cognitive impairment, but a reduced secretion of acetylcholine could aggravate inflammation [39], and acetylcholinesterase inhibitors have been discussed as potential treatment options for cognitive impairment [40,41]. As the effects caused by surgical trauma are relatively well described, it still remains elusive how anesthetics impact on cognitive impairment.

## 3. Anesthetics Related to Cognitive Impairment

The association of the use of special anesthetics regarding cognitive impairment has been studied extensively, but results are sometimes contrary. In general it is estimated that the shorter the duration of action of the anesthetic agent, the shorter the duration of cognitive impairment in the immediate postoperative period [42]. A recent meta-analysis shows that the usage of midazolam, propofol, desflurane and sevoflurane are associated with a higher incidence of delirium compared to dexmedetomidine [43]. It is assumed that the anti-inflammatory and immunomodulatory effects of dexmedetomidine can reduce acute POD [44]. Furthermore, benzodiazepines, opioids, antihistamines and dihydropyridines are associated with an amplified risk of delirium [45]. Several studies show that the use of propofol, dexmedetomidine and fentanyl are associated with a lower risk of cognitive impairment compared to substances such as midazolam, lorazepam, meperidine and morphine [46]. Sedation, especially with benzodiazepine infusions for mechanical ventilation, is associated with a higher risk of delirium; therefore, the German multicenter study: PReOperative Midazolam on OuTcome of Elderly Patients (I-PROMOTE) has been set up to improve the risk-adapted premedication in elderly surgical patients [47]. Of note, the use of propofol in anesthesia is also generally statistically associated with POD [48]. Postoperative memory impairment can also be found in patients who received propofol-based and remifentanil-based anesthesia [49]. However, propofol plus nitrous oxide anesthesia results in better cognition than propofol combined with sevoflurane [50,51]. In other studies, no differences in the incidence of POCD could be shown in patients anesthetized with either xenon, propofol, desflurane or sevoflurane [43,52]. However, as contrary results regarding the impact of inhalational anesthetics on cognitive impairment still exist, a study by Zou YQ et al. was done to assess the effects of different inhalational anesthetics on postoperative cognitive function [52]. As the results from different studies regarding the impact of anesthetics on cognitive function show high variance, the differences may be caused by genetic factors and interindividual differences could be caused by epigenetic factors. However, it is still not well understood how epigenetics impacts this field.

## 4. Epigenetic Mechanisms

It has been proposed earlier that epigenetics might contribute to cognitive impairment for several reasons. However, until 2013, no heritable, genomic indices of persistent POCD or postoperative dementia after surgery had been identified [6].

The risk factors identified as contributing to cognitive impairment are, *inter alia*, exposure to general anesthesia, hypotension, hypoxia, psychoactive drugs and hippocampal inflammation induced by the surgical intervention. In addition, earlier studies indicate that these factors might induce epigenetic dysfunction in the brain, as chromatin remodeling is required for memory-associated gene transcription and expression [19]. Additional epigenetic and environmental factors are supposed to accompany increasing age and play a significant role in the pathogenesis postoperative cognitive impairment, which mainly occurs in postoperative elderly patients and is characterized by troubles in cognition [53].

### Epigenetics Mechanisms Induced by Anesthetics

It has recently been suggested that unknown, novel, epigenetic-related mechanisms could cause anesthetic-induced neuronal toxicity to human neurons, and that epigenetic modifiers, such as DNA-methyltransferase (DNMT) and histone deacetylase inhibitors (HDACis), might be promising targeted therapeutics to mitigate the neurotoxic effects of anesthetics in the developing brain [54].

However, little is known about whether anesthetics induce epigenetic changes in genes related to cognitive impairment, and further research is needed to elucidate the epigenetic mechanisms completely and their role in this field.

Epigenetic modification generally includes variations in DNA methylation, histone acetylation, histone methylation and noncoding RNA (e.g., miRNA, lncRNA, circular RNA) and plays a crucial role in several diseases, but most of the studies have involved in vitro experiments [55]. The epigenetic field of noncoding RNAs has been studied extensively (>1000 results for miRNA AND anesthetics in Medline), whereas the two other most prominent epigenetic modifications of DNA structure, DNA methylation and histone modification, have been only marginally explored in the context of anesthesia. Consequently, we want to focus on DNA and histone modifications in this review.

We summarize the epigenetic effects, detected mainly in animal studies with mice and rats in the last few years, in Table 1 The impact of anesthetics on DNA methylation seems to be mixed. On the one hand, no effect of propofol or desflurane could be detected in the methylation of two important DNA repair genes [56], and it was demonstrated that sevoflurane increased the DNA methylation of the vascular endothelial growth factor promoter region [57]. However, most anesthetics seem to deacetylate histones due to the increased expression of HDACs.

## 5. Postoperative Cognitive Impairment and Epigenetic Mechanisms

We searched Medline with the following search term: “postoperative AND (delirium OR cognitive OR neuroinflammation) AND (epigenetics OR DNA methylation OR histone OR chromatin)” and received 35 results covering the years from 2011 to 2022 (Appendix A). Mainly, results from animal studies were found, while studies of humans were not done. However, we summarize here the main findings of epigenetic mechanisms linked to postoperative cognitive impairment.

As an example, it is proposed that one of the main receptors of sevoflurane, the excitatory gamma-aminobutyric acid A receptor, is linked to the epigenetic mechanism in the developing brain. This can increase neuroapoptosis, elevate serum corticosterone levels and may contribute to sevoflurane-induced long-term cognitive abnormalities [80]. Wu et al. found that when rats are postnatally exposed to isoflurane, a considerable reduction of brain-derived neurotrophic factor (BDNF) in the hippocampus occurs, which is caused by the transcriptional factors-mediated epigenetic modification in the promoter region of *Bdnf* exon. This BDNF reduction leads to an inadequate drive for the synthesis of synaptic proteins, thus, contributing to the hippocampal synaptic and cognitive dysfunction [81].

### 5.1. DNA Methylation

DNA methylation is facilitated by DNA methyltransferases (DNMT1, DNMT3a, DNMT3b) and generally leads to the repression of gene expression through the addition of methyl groups to CpG sites of the gene promoters, which blocks the binding of transcription factors. DNA methylation is one of the most well-described epigenetic modifications, and regulates numerous diverse cellular processes including the silencing of transposable elements, X-chromosome inactivation and tissue-specific gene expression [82].

In the first part of this section, we summarize studies including human subjects. It was demonstrated recently that neurosurgery significantly altered DNA methylation levels of 24 CpG sites of the TNF, IL1B and IL6 gene (Figure 2). In addition, it was shown that the inflammatory methylation index, which was based on the postoperative DNA methylation levels at five selected CpG sites of the genes mentioned above, can be, with moderate accuracy with an area under the curve value of 0.84, a putative detection tool for delirium [83]. Li et al. showed recently that postoperative global hypomethylation of leukocyte DNA is associated with the development of early POCD in elderly patients undergoing hip surgery [84]. A study evaluated the expression and methylation profile of hOGG1 and XRCC1, two important DNA-repair genes, under isoflurane and propofol anesthesia in 40 patients undergoing elective and minimally invasive surgery. Although an increased expression of these genes, depending on surgery time, was detected after isoflurane anesthesia, this was not accompanied by altered methylation in the genes [56].

Sadahiro et al. showed that major surgery induces acute changes in DNA methylation in peripheral blood mononuclear cells associated with immune response pathways that might influence the risk of postoperative complications, including cognitive dysfunction and delirium [85]. Shinozaki et al. demonstrated in a collective of neurosurgical patients that a decrease in the DNA methylation of cytokines gene CpGs in glia cells and blood cells can be seen with aging. As this can affect their expression, additional research is needed to fully elucidate the role of DNA methylation in aging and how it may influence the pathogenesis of delirium [86]. Zhang et al. conducted a study including young nonhuman primates, mice and children and proposed a pathway that can lead to demyelination in the brains of young individuals after sevoflurane anesthesia. They showed that blood folate levels are reduced in children after anesthesia and surgery, which causes a downregulation of thymidylate synthase gene after sevoflurane anesthesia. The primary target of this disrupted folate metabolism is the ERM-like protein (ERMN), whose gene is epigenetically altered after sevoflurane. The increased methylation of the ERMN promotor causes decreased ERMN expression and leads to demyelination in the brain, which causes cognitive impairment (Figure 2). Therefore, anesthesia leads to disrupted folate metabolism and then defects in myelination in the developmental brain, and ERMN is the important target affected by the anesthesia via epigenetic mechanisms [87].

In the context of neurological disorders, DNA methylation in astrocytes appears to be altered, which can impact on both the neuroprotective and neurodegenerative qualities of astrocytes and affects astrocyte differentiation and inflammatory response [88].

Summarizing the effects seen after surgery mostly includes pathways which are related to immune response or neuronal growth (Figure 2).

In the second part, we focus on animal studies that might help us to get a deeper view into the pathomechanisms of cognitive impairment induced by certain anesthetics.

#### 5.1.1. Sevoflurane Effects in Animal Studies

It was recently demonstrated that sevoflurane exposure leads to increased DNMT expression in pyramidal neurons in rats, which causes cognitive impairment and is accompanied by hypermethylation of Reelin genes and BDNF, and subsequently their downregulation, which finally leads to a reduction of dendritic spines in the pyramidal neurons of the hippocampus [89,90] (Figure 2).

Zhong et al. demonstrated that sevoflurane-induced POCD in elderly mice induces compromised levels of global DNA 5′-hydroxymethylcytosine and ubiquitin-like effects with PHD and ring finger domains 2 in the hippocampus and the amygdaloid nucleus, when compared with non-POCD and control mice. In addition, the 5-hydroxymethylcytosin reduction of the promoters of genes associated with development and neural protection, such as BDNF, glial cell-derived neurotrophic factor, acyl-CoA synthetase short chain family member 2 and glucocorticoid receptor, activated the transcription of these genes and could be found in the hippocampus of POCD mice. Contrarily, increased 5-methylcytosine levels on the promoters of glutamate receptor 2 precursor GluR2 and decreased 5-methylcytosine levels of HMGB1 and MMP9 showed an inverse tendency with transcription of these genes [91]. Zhu et al. showed that reduced glucocorticoid receptor expression due to upregulated methylation levels of the promoter region of glucocorticoid receptor exon 17 enhances neuroinflammation and triggers cognitive dysfunction after sevoflurane anesthesia in adult rats [92] (Figure 2). Chastain-Potts et al. showed that sevoflurane exposure of rat pups results in decreased neuronal 5-methycytosine, indicating reduced DNA methylation [93].

#### 5.1.2. Isoflurane Anesthesia Effects in Animal Studies

Klenke et al. recently showed that DNA methylation in the DNA of mouse primary hippocampal neurons after isoflurane exposure alters the methylation of the genes chemokine (C-X-C motif) ligand 12, chemokine (C-X-C motif) ligand 14, GATA binding protein 3, IL11, IL13, and IL4 receptor alpha belonging to inflammatory pathway [94]. Gregoire et al. showed that global DNA methylation is reduced after surgery and that the oral administration of the methyl donor S-adenosylmethionine can attenuate sensory and cognitive impairment associated with nerve injury after isoflurane anesthesia in mice. These effects might be mediated partly by the systemic administration of the methyl donor S-adenosylmethionine over modulation of DNA methylation in the CNS [95].

### 5.2. Histone Acetylation

Histone acetylation and deacetylation are epigenetic processes mediated by histone acetyl transferases and deacetylases (HDACs) [96]. It is generally assumed that the methylation of histone tails can result in either the induction or repression of gene transcription, but acetylation of histone tails is solely associated with active gene expression [96]. Mechanistically, histone deacetylation enhances the binding of histones to DNA and the aggregation of chromosomes, resulting in transcription inhibition, and is connected to heterochromatin, while histone acetylation is connected to transcriptional activation and associated with euchromatin. It is well-known that inflammation and anesthetics can also influence histone acetylation [54,97,98]. Interestingly, HDAC inhibitory activity can be found in drugs with known anti-inflammatory and neuroprotective functions, such as valproic acid [99]. Here, we summarize the effects of distinct anesthetics on histone acetylation. As little is known about the impact of propofol and midazolam on histone acetylation, these drugs may be superior compared to isoflurane and sevoflurane, where several effects on histone acetylation are known.

#### 5.2.1. Propofol-Induced Effects

A recent study indicated that propofol might be involved in the regulation of histone acetylation in cancer development. Holownia et al. showed that propofol regulates histone acetylation in rat astroglial cells by protection against tert-butyl hydroperoxide toxicity [72]. Lin et al. proved that propofol application during early gestation could affect the offspring’s learning and memory by inhibiting histone acetylation [100].

#### 5.2.2. Midazolam-Induced Effects

In 2016, it was demonstrated in rat hippocampus after exposure to a sedative dose of midazolam followed by combined nitrous oxide and isoflurane, that anesthesia causes epigenetic modulations manifested as histone-3 hypoacetylation, with a 25% decrease in histone acetyltransferase activity. This was accompanied by a downregulation of the transcription of BDNF and cellular Finkel-Biskis-Jinkins murine sarcoma virus osteosarcoma oncogene. Hence, the authors postulated that long-term impairments of neuronal development and synaptic communication could be caused by anesthesia-induced epigenetic phenomena [98].

#### 5.2.3. Isoflurane-Induced Effects

The incubation of THP-1 monocytes with isoflurane results in increased HDAC1 and HDAC2 expression and downregulates cytokine expression. These results were shown in primary human peripheral blood monocytes, where HDAC1 and HDAC2 gene silencing caused increased cytokine production and NF-κB nuclear translocation induced by isoflurane pre-exposure and lipopolysaccharide stimulation. These findings showed that anti-inflammatory effects of isoflurane in human monocytes involve the regulation of HDAC1 and HDAC2 [101]. These results are in contrast to those of studies with rats, which showed that isoflurane exposure increased cognitive impairment significantly. Here an upregulation of HDAC 2, reduction of histone acetylation of AcH4K12 and AcH3K9, and increase inflammation (Il-1β, Il-6) (Figure 3) and apoptosis in the hippocampus were shown. Additional impairments of BDNF-tyrosine kinase receptor B (TrkB) signaling and the downstream signaling pathway including phospho-calmodulin-dependent protein kinase and phospho-cAMP response element-binding protein. These findings indicate that isoflurane-induced cognitive dysfunction is associated with the downregulation of the BDNF-TrkB signaling pathway due to declines in chromatin histone acetylation [97]. Hence both above mentioned studies have in common that HDAC2 expression is increased after isoflurane exposure, even if its impact on cytokine expression is contrary. In addition, mice that are neonatally exposed to isoflurane show significant memory impairment and exhibit dysregulated hippocampal H4K12 acetylation and decreased c-Fos expression, which can be attenuated by the HDACi trichostatin A [102]. Jin et al. showed that enhancing the expression of antiapoptotic proteins (Bcl-2, Bcl-xL, xIAP, c-IAP-1, c-IAP-2 and survivin) and improving the acetylation of H3K9 and H4K12 improves the cognitive function of rats after isoflurane anesthesia via the downregulation of the expression of the HDACs HDAC2 and HDAC-3 [103]. Chen et al. demonstrated that isoflurane exposure in aged rats leads to impaired spatial learning and memory, accompanied by dysregulated H3K9 and H4K12 acetylation, which is, in turn, accompanied by reduced BDNF expression and the suppression of the BDNF downstream signaling pathway. Restoring histone acetylation and BDNF signaling suppresses proinflammatory cytokines and the NFκB signaling pathway and attenuates isoflurane-induced cognitive dysfunction [64]. It was demonstrated recently that isoflurane anesthesia increases the expression of HDAC3 protein and decreases levels of dendritic spine density and synaptic plasticity-related proteins in the dorsal hippocampus of aged mice [104].

Zhong et al. showed that neonatal isoflurane exposure-induced memory impairment is associated with the dysregulation of H4K12 acetylation, which may lead to less hippocampal activation and cognitive impairment in mice [105].

#### 5.2.4. Sevoflurane-Induced Effects

Sevoflurane impairs synaptic plasticity and cognitive function in aged mice [106]. It was demonstrated recently that sevoflurane inhibits histone acetylation at H3K18, H3K14, H4K5 and H4K12 (Figure 3). This can induce decreased binding of H3K18 to the promoter of GluN2B and H3K14 to the promoter of the GluN2A gene. In addition, sevoflurane increases ANP32A expression by activating C/EBPβ [107]. All these effects are associated with cognitive impairment.

The cognitive dysfunction after sevoflurane anesthesia can be alleviated by HDACi trichostatin A treatment via inhibition of the overactivation of astrocyte intracellular NF-κB signaling and the release of cytokines [92]. Histone deacetylases modulate cytokine synthesis and release. In line with this, trichostatin A diminishes lipopolysaccharide-induced inflammatory responses in the mouse brain and modulates t cytokine-associated changes in cognitive function, which may be specifically related to reducing HDAC2 and HDAC5 expression [108]. Zhang et al. showed that inhibition of HDAC significantly reverses the sevoflurane-induced decrease in Ac-H3, BDNF, tropomyosin-related kinase B and p-cAMP response element-binding expressions, and reduces the level of apoptosis-related protein-cleaved caspase-3. They conclude that normalizing the hippocampal histone acetylation state might resolve cognitive and synaptic plasticity impairments induced by sevoflurane exposure [109].

### 5.3. Histone Methylation

The role of the methylation of histones facilitated by histone methylases is not yet fully understood. It results in either increased or reduced gene expression, depending on the specific amino acid methylated on the histone tails. It is assumed that H3K4 tri-methylation is associated with transcription increase, whereas H3K9 tri-methylation leads to reduced transcription [110].

Histone methylation has not been studied well in the context of cognitive impairment. Wu et al. showed that histone H3K9 trimethylation downregulates the expression of BDNF in the dorsal hippocampus and impairs memory formation during anesthesia with laparotomy under isoflurane inhalation and surgery in mice [111]. Our group showed recently that propofol decreased H3K4 tri-methylation, and that cholinergic genes bind to this histone [112]. In addition, we demonstrated that midazolam increases the expression of lysine-specific demethylase (KDM1A), which is accompanied by a decreased di-methylation of H3K4 [113]. In addition, a recent network biology approach identified that histone-lysine N-methyltransferase SETDB1, among other genes, is upregulated in patients with POCD [114].

Other histone modifications exist in addition to methylation and acetylation, and include phosphorylation, ubiquitylation and sumoylation. However, not much is known about these modifications in the context of cognitive function, but existing studies indicate that astrocyte-specific histone phosphorylation, ubiquitylation and sumoylation could be important factors to examine in neurodegenerative diseases [115].

## 6. Therapeutic Usage of Epigenetics in Postoperative Cognitive Impairment

All these studies indicate that epigenetic alteration may play a role in anesthesia-induced cognitive impairment. Hence, new therapeutic options for the treatment of POD and POCD might implicate epigenetic drugs, which might be inhibitors of HDACs and methyl donor compounds [116]. As an example, HDACis are small drugs which can cross the BBB, thus slowing the initiation and development of symptoms in animal models of neurodegenerative diseases [117].

Several approaches for the use of HDACis can be found in the literature. Wu et al. demonstrated that the inhibition of HDAC in rats alleviates cognitive impairment after surgery. They showed that the reduction of HDAC2 expression and inhibition of HDAC activity decrease activated microglia levels and increase postsynaptic density 95 protein expression in the hippocampus. In addition, NF-κB-p65 protein expression and nuclear accumulation, as well as the neuroinflammatory response with the proinflammatory cytokines TNF-α and IL-1β in the hippocampus, was reduced by HDAC inhibition [118]. Jia et al. showed that HDAC inhibition by suberoylanilide hydroxamic acid attenuates POCD in aging mice after laparotomy treatment and surgery. HDACi attenuated hippocampus-dependent long-term memory impairments and restored the laparotomy-induced reduction of acetyl-H3 and acetyl-H4 levels in the hippocampus. In addition, it decreased the expression of inducible nitric oxide synthase, cleaved caspase-3, and the N-methyl-D-aspartate receptor-calcium/calmodulin dependent kinase II pathway, and enhanced the gene expression of synapsin 1, BDNF and postsynaptic density 95. Summarizing, these data indicate that the decrease of histone acetylation contributes to cognitive impairment and may be a target to improve its neurological outcome [119]. This is in line with the study of Ji et al., which showed that the cognitive impairments and signaling deficits after isoflurane anesthesia can be rescued by HDACi sodium butyrate. Hence, the epigenetic improvement of BDNF-TrkB signaling may be a promising target for withdrawing isoflurane-induced cognitive impairments [97]. Sun et al. showed in a mouse model that a decreased level of histone acetylation impairs memory, and Hdac2 suppression can reverse microglial activation, hippocampal glutamatergic BDNF and c-Fos expressions, and recover memory deficits [120]. It was demonstrated recently that a HDACi reverses anesthesia-induced (nitrous oxide, Isoflurane, Midazolam) histone-3 hypoacetylation in the immature hippocampus of rats and restores the anesthesia-induced alteration of inhibitory synaptic transmission in the developing rat hippocampus [121]. Hence, HDACis might be promising novel therapeutic options for postoperative cognitive impairment.

5-azacytidine or azacitidine (AzaC) is one of the most widely studied DNMT inhibitors, and is used mainly for managing myelodysplastic syndrome, in which it functions as a methyl donor-remover in vitro to weaken the gene silencing effect upon methylation [122]. It was demonstrated that the inhibition of DNMT activity with AzaC increases the protein expression of BDNF in hippocampal neurons [123]. This could enhance cognitive function. In addition, nutrition and dietary compounds, such as vitamin B, folate, and methionine, are known to be effective for the modulation of DNA methylation [122]. Furthermore, dietary compounds can modulate the activity of DNA-methytransferases protein-related methylation, as they can inhibit DNMTs [124].

## 7. Discussion

Anesthesia is usually considered to safely induce a reversible unconscious state, allowing surgery to be performed without pain. However, an increasing number of studies, especially preclinical ones, demonstrate that general anesthetics can cause long-term cognitive impairment [20]. Since a causal relationship is currently unknown, we wrote this review to evaluate any possible epigenetic mechanisms of how an anesthetic drug could contribute to postoperative cognitive impairment. Hence, we summarize the role of epigenetic modifications in the development of cognitive impairment after anesthesia. The role of epigenetics is still a novel field, which has to be explored further. As mentioned above, the studies presented here are mostly preclinical studies particularly from animals, which have to be verified by the examination of human subjects. We focus in the first part of this review on the pathomechanisms, which induce cognitive impairment after anesthesia, and it becomes clear that neuroinflammation has a significant impact on its genesis [23]. However, we have to keep in mind that cognitive impairment after anesthesia may have multiple causes, which include pre-operative general and cognitive health, medical comorbidities like hypotension, hypoxia, hospital admission for any cause, exposure to general anesthesia and psychoactive drugs [19], and that neuroinflammation may only be a small piece of the whole picture. Secondly, we show that anesthetic alters the expression of histone and DNA-modifying enzymes. With a deeper view into the mechanism of DNA methylation in the context of cognitive impairment, it becomes clear that anesthesia is capable of both increasing and decreasing DNA methylation. Interestingly, most preclinical studies show that the methylation of genes related to inflammation is decreased, which causes an enhanced neuroinflammation, and the methylation of genes related to neuronal growth is increased, which hinders neuronal development. Some studies focus on cholinergic genes in this context with mixed results [13,27,37,38,41,112,113,125,126,127,128,129,130]. Histone modifications in the anesthesia-induced cognitive impairment focus mainly on histone acetylation and seldom on methylation, but other important modifications are absent, such as histone phosphorylation. Numerous studies indicate that anesthesia causes an increase in HDACs, which is associated with the increased NF-kB pathway, and the release of cytokines and neuroinflammation. Interestingly, the inhibition of histone acetylation leads to decreased BDNF expression, which can cause a deficit in neuronal growth.

We have to think about the clinical implications of the described results. In clinical practice, explaining that an epigenetic change can induce post-anesthesia delirium might be hard to believe because of the chronology from direct damage to symptom onset and the time that would be expected to exist between the epimutation or epigenetic change and its translation into the clinical phenotype. This seems especially troubling when considering genes involved in neurodevelopment and neurogenesis. However, genes related to neurogenesis, such as BDNF, as mentioned here, seem to play a role in age related cognitive impairment as well [131]. Comparing the findings from this review to other causes of delirium makes clear that in delirium from other causes mostly biomarkers regarding inflammatory pathways are involved (e.g., IL-6, CRP, Corsitol, TNF-alpha) [24]. In addition, some studies report that non-degenerative cognitive impairment can be related to biomarkers such as t-tau, p-tau181 or S100B, while in others this is only reported for degenerative cognitive impairment [132].

Much more future research is needed. This means that more systematic investigations are needed to determine the influence of different anesthetics depending on cell populations and in patients of different ages to analyze the precise effects. In addition, it should also be noted that large randomized prospective trials are currently missing to explore the exact epigenetic mechanisms and analyze possible influences on the patient’s outcome. This will help us to translate the findings from animal studies to clinical practice and can determine if anesthetics directly induce long-lasting epigenetic effects, which impact on genes of neuroinflammation and neurodevelopment. Furthermore, before epigenetically drugs can be used in clinical settings, it has to be clarified, how they can be used in a targeted approach [133]. However, the success of therapeutic intervention using epigenetic modifiers, such as DNMT inhibitors and HDACis, implies that the epigenetic drugs may be promising as potential targeted therapies to mitigate neurotoxic effects of anesthetics. As cognitive impairment is closely related to neuroinflammation, the successes of therapeutic intervention in chronic inflammatory diseases using epigenetic modifiers, such as HDACis and inhibitors of DNA methylation, suggest that the epigenetic intervention will be very promising for postoperative cognitive impairment in the near future [134]. However, one important challenge in the epigenetic intervention of cognitive impairment is the identification of inhibitors for specific histone-modifying enzymes or specific DNA methylation sites [135].

## 8. Conclusions

In conclusion, epigenetic mechanisms seem to play a critical role in the development of POCD. Although the data are sometimes heterogeneous, it seems that epigenetic modifications effect neuroinflammation, neuronal development and the secretion of neurotransmitters. Surgery and anesthesia might trigger the epigenetic dysregulation of cognition-associated genes, which alter the hippocampal gene expression program associated with memory consolidation and, consequently, leads to cognitive impairment following surgery. The restoration of abnormal chromatin remodeling and DNA methylation might be a viable approach to therapeutic intervention in postoperative cognitive impairment.

## Figures and Tables

**Figure 1 cells-11-02954-f001:**
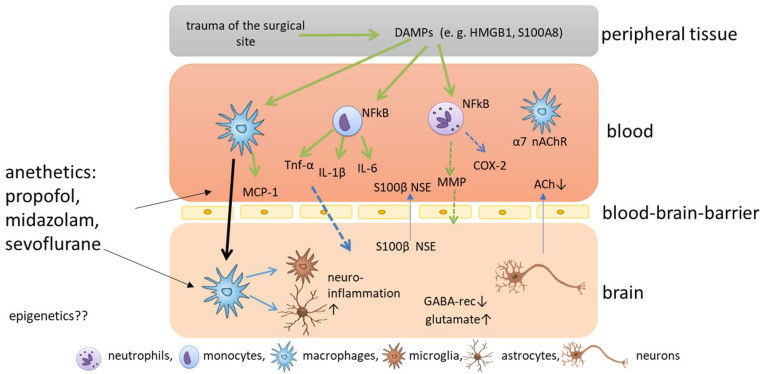
Schematic overview of the molecular mechanisms of postoperative cognitive impairment. Trauma of the surgical site causes the release of damage associated patterns (DAMPS), which are released into the blood where nuclear factor kappa B is activated in immune cells, such as neutrophils, monocytes and macrophages, which release cytokines and chemokines. The release of matrix metalloproteinases (MMPs) and cyclooxygenase 2 (COX2) cause permeabilization of the blood-brain barrier and pro-inflammatory cytokines and macrophages enter the brain. Here, microglia are activated and neuroinflammation is amplified.

**Figure 2 cells-11-02954-f002:**
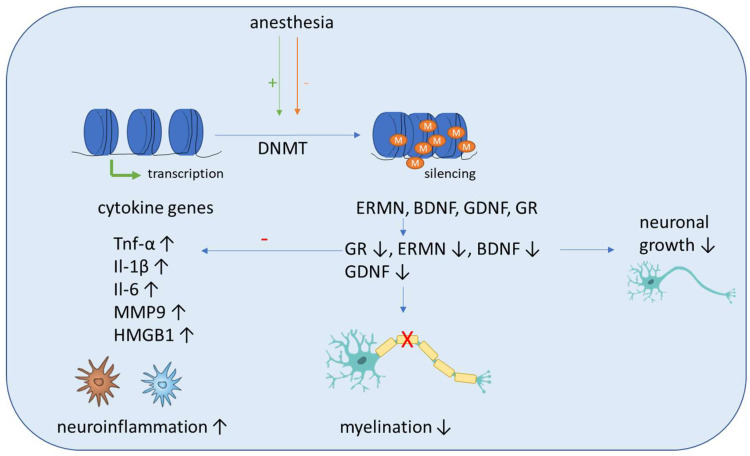
Schematic overview of the molecular mechanisms induced by anesthesia on DNA methylation. Anesthesia can cause both an increase and decrease of DNA-methyltransferase (DNMT) activity. The methylation of inflammatory genes (Tnf-α, IL-1β, Il-6, MMP9, HMGB1) is generally decreased, which leads to increased expression and aggravates neuroinflammation. Anti-inflammation is decreased by decreased expression of the anti-inflammatory transcription factor glucocorticoid receptor (GR). In addition, the methylation of genes associated with neuronal growth (BDNF, GDNF), differentiation and survival of existing neurons (ERMN) is increased, which reduces their expression and causes cognitive impairment due to demyelination and the reduced growth of neurons.

**Figure 3 cells-11-02954-f003:**
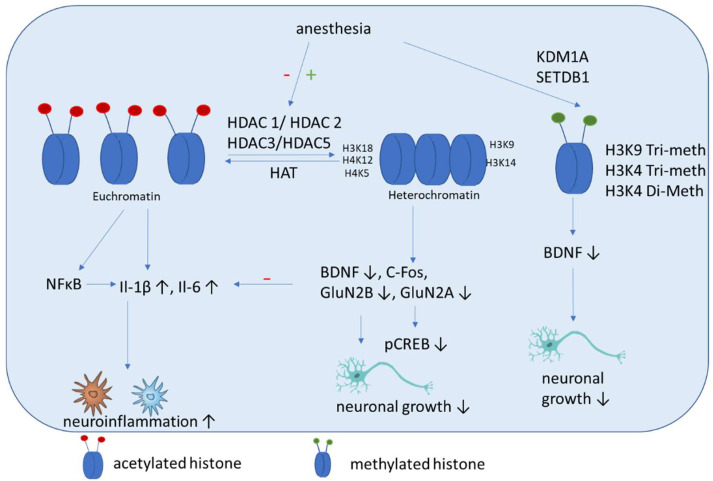
Schematic overview of the molecular mechanisms induced by anesthesia on histone modifications. Anesthesia can cause both a decrease of histone acetylation and an increase of HDAC activity. Histone deacetylation mainly causes an increase of inflammatory cytokines and a decrease of factors associated with neuronal growth, differentiation and survival of existing neurons. In addition, methylation (SETDB1) and demethylation (KDM1A) are induced by anesthesia, which can cause a decrease of BDNF.

**Table 1 cells-11-02954-t001:** Overview of epigenetic modifications caused by anesthetics and their possible impact on cognitive impairment (n.a. not analyzed so far).

Anesthetic	DNA Methylation	Histone Modification	Association with Cognitive POD/POCD
Desflurane	n. a.	n. a.	
Isoflurane	DNMT1 ↓ [58] Methylation ↓ Cxcl12-methyl. ↑ methylation levels at the D-loop ↓ [59] Ephrin pathway gene methylation ↑ [60]	Acetylation ↓ [61] HDAC2 ↑ HDAC4 ↑ [62] HDAC8 ↑ [63] GLT-1 act. ↓ H3K9 and H4K12 acetylation ↓ [64] BDNF ↓ [65] H3K9 trimethy ↑ HDAC ↑ and H3-ac ↓ H4-ac ↓ neuroligin 1 ↓ [66]	Cognitive impairment in rats and mice
Sevoflurane	Methylation in hippocampus ↑ Kcc2 gene methylation ≠ [67] ↑ VEGF-methyl. ↑ HMGB1-methyl. ↓ MMP9-methyl. ↓ GLUR2-methyl. ↑ DNMT ↑ BDNF-methyl. ↑	HDAC3 ↑ [68] HDAC6 ↑ [69] HDAC8 ↑ acetyl-H3K9 ↓ and acetyl-H3K14 ↓ H4-acetyl. ↓ histone acetyltransferase (HAT) ↓	Cognitive impairment in rats
Midazolame	n.a.	n.a. alone (comination with isoflurane)	
Propofol	XRCC1-methyl-. = [70] hOGG1-methyl. = DNTM3A ↓ DNMT3B ↓ [71]	HDAC2 ↑ [72] H3-actetyl. ↓ H4-acetyl. ↓ H3K27 trimethy. ↓ HDAC1 ↓	n.a.
Ketamine	COX2-methyl. ↓ [73] Hypermethylation hyper-methylation of Bdnf gene[74]	HDAC ↑ HDAC5 ↑ HDAC6 ↓ [75] H3K27m3 ↓ H3K36m3 ↓	
pentobarbital sodium		HDAC2 ↑ [76]	Cognitive dysfunction in mice
fentanyl	DNMT1 ↓ DNMT3A ↓ DNMT3B ↓ [77]	HDAC5 ↓ [78] Sirtuin 2 [79]	

## Data Availability

Not applicable.

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
