# Peer review of "Epigenetic Mechanisms of Postoperative Cognitive Impairment Induced by Anesthesia and Neuroinflammation"

_cells, 2022, doi:10.3390/cells11192954_

Round 1

Reviewer 1 Report

First of all, I congratulate and thank the authors for choosing a novel approach to the subject of post-anesthesia cognitive impairment. I believe that on the whole it is a quality work but I have some comments/suggestions for improvement that concern rather the more clinical part of the work, I detail them below: 

1. Explain better the concept of delirium or acute confusional syndrome vs cognitive impairment without confusional syndrome. Both are associated with cognitive impairment, the duration, onset of symptoms and the existence or not of accompanying symptoms is what varies. Also differentiate between the terms mild cognitive impairment and dementia, they are not the same. 
Mention other causes of delirium (hospital admission for any cause, infections...) and cognitive impairment. 
What is the neuropsychological profile of cognitive impairment in case of delirium associated with anesthesia? The same as in other causes of delirium? And in post-anesthesia cognitive impairment without delirium? 
What are the predictors (demographic, clinical variables...) that predict progression of delirium to non-reversible cognitive impairment? 

2. About the figures: In figure 2 modify the upper part. The effect of DNMT on methylation and of methylation on transcription can be expressed in a schematic and more graphic way. In figure 3 try to modify the upper part also, it is not clear the graphical representation of chromatin and the effect of acetylation. 

3. Value to create as supplementary material all the search strategies and search engines used for the realization of this narrative review.  You mention the search strategy in some sections, but it would be better to have a separate and more detailed section. It can provide information on the "quality" of the review, the "thoroughness" of the search, which is always welcome. 

4. What is rather described is the possible association between epigenetic changes and anesthesia. It focuses on genes involved in the causes of cognitive impairment but beware of making causal associations. In clinical practice, explaining that an epigenetic change can induce post-anesthesia delirium is hard to believe because of the chronology from direct damage to symptom onset and the time that would be expected to exist between the epimutation or epigenetic change and its translation into the clinical phenotype (especially in the case of genes involved in neurodevelopment, neurogenesis...). 
I would recommend in the discussion section to compare the findings with other causes of delirium and cognitive impairment (degenerative and non-degenerative). I think it would be of great interest. 

5. Evaluate creating a future research section, perhaps accompanied by clinically relevant and unanswered questions. I think it would be of interest. 

6. Consider reducing the text of the description of the effect of each drug. I think that much of the content can go to table format, now it is tedious to read and not as attractive as it could be.

I believe that if the authors are able to answer these questions it could be evaluable for publication.

Author Response

Reviewer 1

First of all, I congratulate and thank the authors for choosing a novel approach to the subject of post-anesthesia cognitive impairment. I believe that on the whole it is a quality work but I have some comments/suggestions for improvement that concern rather the more clinical part of the work, I detail them below:

  1. Explain better the concept of delirium or acute confusional syndrome vs cognitive impairment without confusional syndrome. Both are associated with cognitive impairment, the duration, onset of symptoms and the existence or not of accompanying symptoms is what varies. Also differentiate between the terms mild cognitive impairment and dementia, they are not the same.

Mention other causes of delirium (hospital admission for any cause, infections...) and cognitive impairment.

What is the neuropsychological profile of cognitive impairment in case of delirium associated with anesthesia? The same as in other causes of delirium? And in post-anesthesia cognitive impairment without delirium?

What are the predictors (demographic, clinical variables...) that predict progression of delirium to non-reversible cognitive impairment?

In response: Thank you very much for your valuable comments. We now tried to explain the concepts better and added explanations in the introduction on pages one and two.

Line 34 ff:

Delirium or acute confusional syndrome is characterized by impaired orientation, diminished attention, and aberrant perception. It is defined as a syndrome characterized by a fluctuating cognitive impairment of acute onset that resolves within days of a surgical procedure. The pathogenesis is multifactorial and it frequently appears in elderly patients admitted to general hospitals [5]. The contrast to cognitive impairment without confusional syndrome is that it varies in its duration. It can last 3 months or longer [6]. In this context we also have to mention mild cognitive impairment and dementia, which are not the same although both are characterized by objective evidence of cognitive impairment, the main difference is that in dementia, more than one cognitive domain is invariably involved and substantial interference with daily life is evident [7]. In addition, dementia is an irreversible state, which distinguishes it from the here described phenomena. The relationship between POD and POCD is not fully understood and overlap and divergence are currently being discussed [2,8]. Delirium might contribute to POCD or long term cognitive impairment due to several reasons, one is that patients with delirium are often incapable of maintaining hydration and nourishment, even with good nursing care, in addition sleep deprivation might occur and cause neuronal damage, in addition drug toxicity might occur, as an example the use of benzodazepines can be associated with cognitive impairment [9].

Line 65 ff:

The pathophysiology of the syndromes is still not well understood, however it is assumed that in delirium an acute change of neurotransmitter levels occur, while long term cognitive impairment is accompanied by chronic neurodegeneration and triggered by neurotoxicity [9]-.

Line 70 ff:   

However other risk factors that might contribute to cognitive impairment have to be mentioned. They include pre-operative general and cognitive health, medical comorbidities like hypotension, hypoxia, hospital admission for any cause, exposure to general anesthesia and psychoactive drugs [16]. In the postsurgical period neuroinflammation caused by surgery is considered a major contributor to the development and ageing, atherosclerosis and neurodegenerative diseases may lead to an increased systemic and cerebral inflammatory response to surgery. In addition  postoperative complications, such as infection, pulmonary complications and hypoxaemia may occur and aggravate the postoperative inflammatory reaction [17]

  1. About the figures: In figure 2 modify the upper part. The effect of DNMT on methylation and of methylation on transcription can be expressed in a schematic and more graphic way.

In figure 3 try to modify the upper part also, it is not clear the graphical representation of chromatin and the effect of acetylation.

In response: Thank you very much for your hints. Figures 2 and 3 were now modified. We excuse for confusion in figure 3, but we hope that everything is presented correctly now.

  1. Value to create as supplementary material all the search strategies and search engines used for the realization of this narrative review. You mention the search strategy in some sections, but it would be better to have a separate and more detailed section. It can provide information on the "quality" of the review, the "thoroughness" of the search, which is always welcome.

In response: Thank you again for this valuable comment. We now added a supplemental figure at the end of the manuscript. This can be found in lines 574 ff.

  1. What is rather described is the possible association between epigenetic changes and anesthesia. It focuses on genes involved in the causes of cognitive impairment but beware of making causal associations. In clinical practice, explaining that an epigenetic change can induce post-anesthesia delirium is hard to believe because of the chronology from direct damage to symptom onset and the time that would be expected to exist between the epimutation or epigenetic change and its translation into the clinical phenotype (especially in the case of genes involved in neurodevelopment, neurogenesis...).

I would recommend in the discussion section to compare the findings with other causes of delirium and cognitive impairment (degenerative and non-degenerative). I think it would be of great interest.

In response: Thank you very much for this suggestion. We now included a paragraph in the discussion, which addresses these points on lines 523 ff:

We have to think about the clinical implications of the described results. In clinical practice, explaining that an epigenetic change can induce post-anesthesia delirium might be hard to believe because of the chronology from direct damage to symptom onset and the time that would be expected to exist between the epimutation or epigenetic change and its translation into the clinical phenotype. This seems especially different when considering genes involved in neurodevelopment and neurogenesis. However, genes related to neurogenesis like BDNF, which was mentioned here seems to play a role in age related cognitive impairment as well [133]. Comparing the findings from this review to other causes of delirium makes clear that in delirium of other causes mostly biomarkers regarding inflammatory pathways are involved (e.g. IL-6, CRP, Corsitol, TNF-alpha) [24]. In addition, some studies report that non-degenerative cognitive impairment can be related to biomarkers like t-tau, p-tau181 or S100B, while in other this is only reported for degenerative cognitive impairment [134]. 

  1. Evaluate creating a future research section, perhaps accompanied by clinically relevant and unanswered questions. I think it would be of interest.

Thank you for this outstanding comment. We now rewrote and supplemented the last part of the discussion. The text in line 536ff now reads:

Much more future research is needed. This means that, more systematic investigations are needed to determine the influence of different anesthetics depending on cell populations and in patients of different ages to analyze the precise effects. In addition, it should also be noted that large randomized prospective trials are currently missing to explore the exact epigenetic mechanisms and analyze possible influences on the patient’s outcome. This will help us to translate the findings from animal studies to the clinical practice and can answer the question if anesthetics indeed directly induce long-lasting epigenetic effects, which impact on genes of neuroinflammation and neurodevelopment. Furthermore, before epigenetically drugs can be used in clinical settings it has to be clarified, how they can be used in a targeted approach [135].

  1. Consider reducing the text of the description of the effect of each drug. I think that much of the content can go to table format, now it is tedious to read and not as attractive as it could be.

In response: Again, we would like to thank you for the effort and thought you put into improving the manuscript. However, as some results regarding certain anesthetics are contrary, we decided that explaining this in a text is easier to follow for the readers. In addition, as one is only interested in the big picture, one can view the illustrations. Though if, despite these remarks, you are still convinced that it is better to make a table, we will do so in a second round of review.

I believe that if the authors are able to answer these questions it could be evaluable for publication.

Reviewer 2 Report

An old review of 2013 by Wang and colleagues have already presented epigenetics mechanisms as possible explanation of postsurgical cognitive impairment in older people, proposing it as a possible target for ameliorating impairment in patients. Here, only neuroinflammation is presented as a possible cause inducing epigenetics changes.

Other factors including exposure to general anesthesia, hypotension, hypoxia, use of psychoactive drugs in geriatric patients should be reported together with hippocampal inflammation induced by the surgical intervention.

2) a scheme summarizing the current hypotheses should be useful to depict the aim of this review

3) is it reported a correlation among anesthesia duration and its effects in impairment?

4) error in formatting table 1: the table should be entirely contained in the same page

5) why do you explore only period 2011-2022 for literature about this topic?

You should clarify this point.

Author Response

Reviewer 2

An old review of 2013 by Wang and colleagues have already presented epigenetics mechanisms as possible explanation of postsurgical cognitive impairment in older people, proposing it as a possible target for ameliorating impairment in patients. Here, only neuroinflammation is presented as a possible cause inducing epigenetics changes.

Other factors including exposure to general anesthesia, hypotension, hypoxia, use of psychoactive drugs in geriatric patients should be reported together with hippocampal inflammation induced by the surgical intervention.

In response: Thank you very much for your valuable comment. We agree that we should discuss other risk factors more extensively and added the following paragraph on lines 52 ff:

However other risk factors that might contribute to cognitive impairment have to be mentioned. They include pre-operative general and cognitive health, medical comorbidities like hypotension, hypoxia, exposure to general anesthesia and psychoactive drugs [15]. In the postsurgical period neuroinflammation caused by surgery is considered a major contributor to the development and ageing, atherosclerosis and neurodegenerative diseases may lead to an increased systemic and cerebral inflammatory response to surgery. In addition  postoperative complications, such as infection, pulmonary complications and hypoxaemia may occur and aggravate the postoperative inflammatory reaction [16].

2) a scheme summarizing the current hypotheses should be useful to depict the aim of this review

In response: Thank you very much for this valuable comment. We now included a scheme with the main hypotheses in line 70:

Main hypotheses of the review:

-        Cognitive impairment is a common problem after surgery

-        Epigenetic dysregulation may play a role in its development

-        Anesthetics alter expression of DNA and histone modifying enzymes

-        This causes and altered epigenetic profile of genes involved in neuronal maintenance and development and neuroinflammation

-        Epigenetic drugs might be a novel therapeutic option for the treatment of postoperative cognitive impairment

3) is it reported a correlation among anesthesia duration and its effects in impairment?

In response: Yes, the duration seems to have impact on the severity of cognitive impairment. We now included in line 127 ff.:

In general it is estimated that the shorter the duration of action of the anesthetic agent, the shorter the duration of cognitive impairment in the immediate postoperative period [38].

4) error in formatting table 1: the table should be entirely contained in the same page

In response: Thank you very much for this helpful comment. A table break was now included.

5) why do you explore only period 2011-2022 for literature about this topic?

In response: Thank you again for your valuable comments, which improved our manuscript. We now added a supplemental figure in line 574 ff. to clarify this point. When using the search term: “postoperative AND (delirium OR cognitive OR neuroinflammation) AND (epigenetics OR DNA methylation OR histone OR chromatin)” only relevant results of the years 2011 to 2022 appear. I hope this clarifies the point.

You should clarify this point.

Reviewer 3 Report

Broader comments:

Authors have done a good work on summarizing the recent research and developments in combination to connect epigenetics to neuro inflammation.

There are multiple disease events that are cover here (AD, Neuro development and cancer) each disease have different molecular and epigenetics phenotypes. Use of Anesthesia during the operative condition may have very large effect on epigenetic change, correlating the single molecule and to neuro inflammation is a too much of a vacuum. 

After Histone Acetylation: at end of 5.2 section It's good have summary of each drug and their effect on epigenetic and which of them have minimal side effects. 

In section 5.2.3 - for Isoflurane - you have referenced 96 and 97; at line no 339 - can you briefly mention about the difference in the outcome of each study.

Specific comments

1. Line 134-135 : reference the main article where the work done 

2. Line 137-138 : Reference no 45 - does not talk about the fact you have mentioned (wrong citation)

3. Table 1 :  table talks about DNA methylation for Sevoflurane -you have mentioned m6A methylation, which is a RNA methylation 

In the table one N.A refers to study not done or NO change in methylation?

4. Line 144 : '' While studies of human are mostly missing'' - Here it should be done are not done (mostly missing is inappropriate) 

5. Line 277 : What is IL4ra?

6. Line 289 : Anti inflammatory spelling mistake

7.  All inflammatory gene  representation is not uniform

8. Line no 302-302: Add more relevant references

9. Line no- cite main study and not the reference study

10. Line no 331-336 : I suggest to split the sentence, as its too many events 

11. section 5.3 - Line no 385 - reference is not right!

Author Response

Broader comments:

Authors have done a good work on summarizing the recent research and developments in combination to connect epigenetics to neuro inflammation.

In response: Thank you very much for your valuable comments and for generally evaluating the work positively.

There are multiple disease events that are cover here (AD, Neuro development and cancer) each disease have different molecular and epigenetics phenotypes. Use of Anesthesia during the operative condition may have very large effect on epigenetic change, correlating the single molecule and to neuro inflammation is a too much of a vacuum. 

In response:  Thank you very much for this valuable comment. We now added more information in the discussion section on line 512 ff:

However, we have to keep in mind that cognitive impairment after anesthesia might have multiple causes which include pre-operative general and cognitive health, medical comorbidities like hypotension, hypoxia, hospital admission for any cause, exposure to general anesthesia and psychoactive drugs [19] and that neuroinflammation may only be a small piece of the whole picture.

After Histone Acetylation: at end of 5.2 section It's good have summary of each drug and their effect on epigenetic and which of them have minimal side effects. 

In response: Thank you very much for this suggestion. We now added the following sentence at the end of section 5.2: line 307 ff:

As only little is known about the impact of propofol and midazolam on histone acetylation, these drugs might be superior compared to isoflurane and sevoflurane, where several effects on histone acetylation are known.

In section 5.2.3 - for Isoflurane - you have referenced 96 and 97; at line no 339 - can you briefly mention about the difference in the outcome of each study.

In response: Thank you for this comment, we now included the following sentence in line 343 ff.:

Hence both above mentioned studies have in common that HDAC2 expression is increased after isoflurane exposure, even if its impact on cytokine expression is contrary.

Specific comments

  1. Line 134-135 : reference the main article where the work done 

We excuse this mistake!

  1. Line 137-138 : Reference no 45 - does not talk about the fact you have mentioned (wrong citation)

We excuse this mistake!

  1. Table 1 :  table talks about DNA methylation for Sevoflurane -you have mentioned m6A methylation, which is a RNA methylation 

We excuse this mistake, which was corrected now.

In the table one N.A refers to study not done or NO change in methylation?

In response: This refers to not analyzed so far.

  1. Line 144 : '' While studies of human are mostly missing'' - Here it should be done are not done (mostly missingis inappropriate) 

We corrected it.

  1. Line 277 : What is IL4ra?

We included it.

  1. Line 289 : Anti inflammatory spelling mistake

We corrected it.

  1. All inflammatory gene  representation is not uniform

I am sorry but I don’t understand what you mean.

  1. Line no 302-302: Add more relevant references

Some more references

  1. Line no- cite main study and not the reference study

Sorry, but there is no line number.

  1. Line no 331-336 : I suggest to split the sentence, as its too many events 

The sentence was no splitted.

  1. section 5.3 - Line no 385 - reference is not right!

We changed this.

Again, thank you for your valuable comments, which improved our manuscript.